# Origins of Sulfate in Groundwater and Surface Water of the Rio Grande Floodplain, Texas, USA and Chihuahua, Mexico

Christopher Eastoe [1,*], Barry Hibbs [2], Mercedes Merino [3] and Jason Dadakis [4]

1   Department of Geosciences, University of Arizona, Tucson, AZ 85719, USA
2   Department of Geological Sciences, California State University-Los Angeles, Los Angeles, CA 90032, USA; bhibbs@exchange.calstatela.edu
3   Engineering Geologist, California Department of Transportation, 100 S Main Street, Los Angeles, CA 90012, USA; mercedes.merino@dot.ca.gov
4   Orange County Water District, 18700 Ward Street, Fountain Valley, CA 92708, USA; jdadakis@ocwd.com
*   Correspondence: eastoe@email.arizona.edu; Tel.: +1-520-204-0684

**Abstract:** Sulfate isotopes ($\delta^{34}$S, $\delta^{18}O_{SO4}$) interpreted in conjunction with sulfate concentrations show that sulfate of both agricultural and geologic sources is present in groundwater and surface water in the Rio Grande flood plain within the Hueco Bolsón. From previous studies, water isotopes ($\delta^2$H, $\delta^{18}$O) in the study area indicate groundwater age relative to dam construction upstream. Surface water entering the Hueco Bolsón contains a mixture of soil-amendment sulfate and sulfate from deep-basin groundwater seeps at the terminus of Mesilla Valley. In the shallow Rio Grande alluvial aquifer within the Hueco Bolsón, ranges of $\delta^{34}$S in pre-dam (+2 to +9‰) and post-dam (0 to +6‰) groundwater overlap; the range for post-dam water coincides with common high-sulfate soil amendments used in the area. Most post-dam groundwater, including discharge into agricultural drains, has higher sulfate than pre-dam groundwater. In surface water downstream of Fabens, high-$\delta^{34}$S (>+10‰) sulfate, resembling Middle Permian gypsum, mixes with sulfate from upstream sources and agriculture. The high-$\delta^{34}$S sulfate probably represents discharge from the regional Hueco Bolsón aquifer. In surface water downstream of Fort Hancock, soil-amendment sulfate predominates, probably representing discharge from the Rio Grande alluvial aquifer near the basin terminus. The $\delta^{18}O_{SO4}$ dataset is consistent with sulfate origins determined from the larger $\delta^{34}$S dataset.

**Keywords:** groundwater; surface water; salinity; sulfate; stable isotopes; Texas; Chihuahua

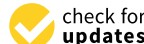



## 1. Introduction

Sulfate isotopes ($\delta^{34}$S, $\delta^{18}O_{SO4}$) are useful in determining sources of sulfate dissolved in groundwater and surface water. Large isotope distinctions occur between sulfate of marine and evaporitic origin, sulfate derived from oxidation of sulfide, sulfate partially affected by bacterial reduction and human-made sulfate from sources such as fertilizer and detergents. A recent review [1] of research using sulfate isotopes in groundwater included studies in which the effects of agricultural pollution on groundwater and watersheds have been resolved using sulfate isotopes [2–6]. In southwestern North America, sulfate isotopes have been used to identify seawater salinization of the Costa de Hermosillo coastal aquifer, Sonora [7], to identify groundwater flow paths in Tucson Basin, Arizona [8,9], to constrain the origin of solutes in Mesilla Valley, New Mexico [10], to distinguish marine evaporite and igneous sulfate sources in native saline groundwater of the Hueco Bolsón [11], to examine sulfate chemistry in leachate from sulfur trapped from flue gas [12] and to quantify relative contributions of evaporitic sulfate and acid rock drainage to Sonoita Creek, Arizona [13].

The floodplain of the upper Rio Grande (in Mexico called the Río Bravo) in New Mexico, northern Chihuahua and west Texas (Figure 1) is a productive agricultural area located within a zone of arid climate. Local agriculture is reliant on irrigation with surface water from the Rio Grande, supplemented with groundwater at times of drought. Salinity

and Cl/SO$_4$ clearly increase in Rio Grande surface water as the river flows through the region [14]. Groundwater in a broad area of the floodplain near the Rio Grande, 3 km southeast of Fabens, Texas, has elevated salinity relative to neighboring parts of the floodplain [15–17].

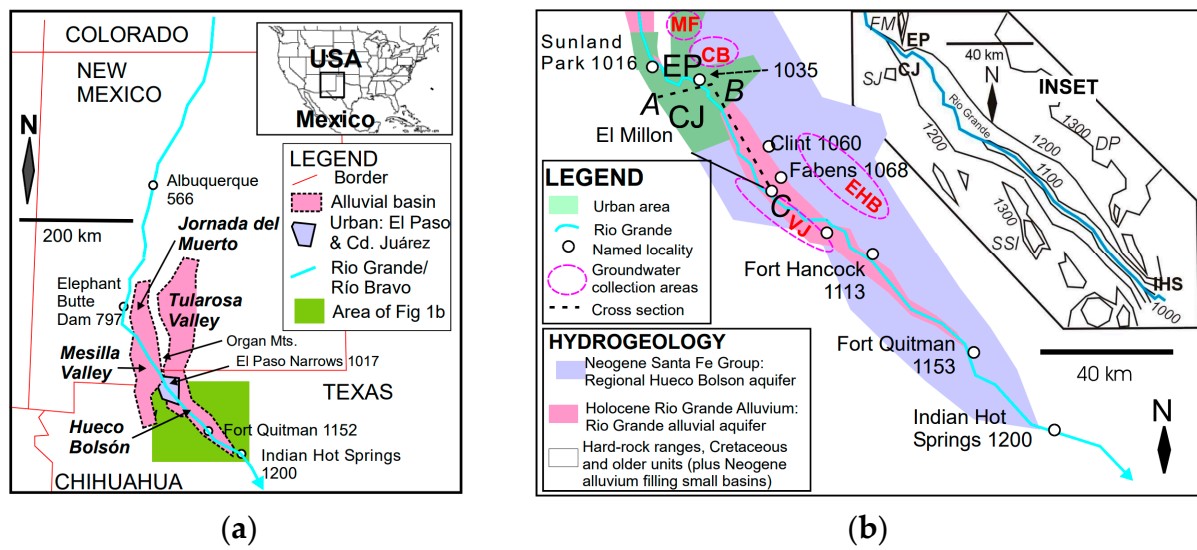

**(a)**　　　　　　　　　　　　　　　　　　　**(b)**

**Figure 1.** (**a**) Regional location map with an inset showing the location of the figure in North America. (**b**) Map of the southern part of the Hueco Bolsón. Grounwater collection areas are shown thus: CB = central basin; MF = mountain front; VJ = Valle de Juárez; EHB = east Hueco Bolsón. River locations shown with km from the source. EP = El Paso, CJ = Ciudad Juárez, IHS = Indian Hot Springs, ABC = cross-section in Figure 2.

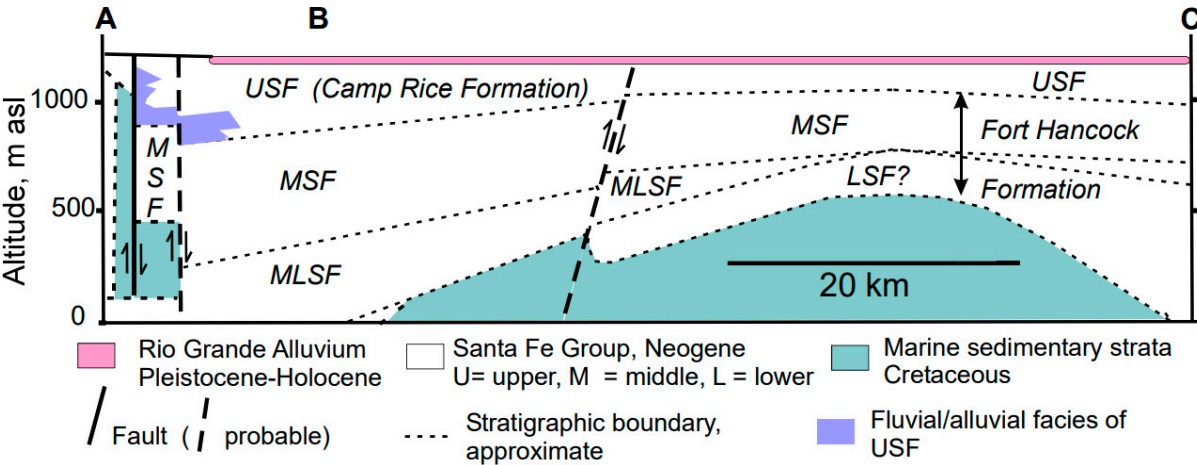

**Figure 2.** Geologic cross-section from Sierra de Juárez to Fabens (see Figure 1). USF = upper Santa Fe Group; MSF = middle Santa Fe Group; LSF = lower Santa Fe Group; MLSF = middle or lower Santa Fe Group. A, B and C are points shown in Figure 1.

Sources of salinity in the Rio Grande floodplain were studied using parameters including major ion geochemistry and stable isotopes ($\delta^{34}$S, $\delta^{18}$O$_{SO4}$) in sulfate [10]. The principal focus of that study was the Mesilla Valley upstream of El Paso (Figure 1), but data were also presented for the section of the floodplain in the Hueco Bolsón, the basin downstream of El Paso. The authors concluded that agriculture was principally responsible for salinity increase in surface water of the floodplain. That conclusion may be inadequate for the Hueco Bolsón where few data were available.

We have assembled additional data from the Hueco Bolsón in order to undertake a new examination of sulfate sources in the agricultural areas of the basin. A large dataset

of $\delta^{34}S$ values paired with sulfate concentrations [$SO_4$] is available from previous studies in the Hueco Bolson [11,18,19]. To this have been added unpublished data ($\delta^{34}S$, $\delta^{18}O_{SO4}$, [$SO_4$]) for Rio Grande surface water collected in 2001 [14] and by the authors in 2001–2005, and for groundwater from the Valle de Juárez and a few samples of soil amendments (Supplementary Material: Table S1). In addition, this study uses an extensive dataset [10] of $\delta^{34}S$ and $\delta^{18}O_{SO4}$ values for surface water from the Rio Grande in Mesilla Valley just upstream of the Hueco Bolsón and for soil amendments used by farmers in the Rio Grande floodplain within the Hueco Bolsón. The data obtained by the authors represent a juxtaposition of agricultural and natural geological effects in 2001–2005 that has subsequently been modified by expanding urban development with diversion of river water for municipal supply, and increasing distribution of reclaimed municipal wastewater for irrigation at a time of intensifying drought. Comparable river samples are now difficult to collect because access to the river channel is limited by a border wall in El Paso County.

Our principal aim is to improve the understanding of sulfate sources in the agricultural areas of the Hueco Bolsón. We approach this topic with a thorough examination of a large dataset of chloride and sulfate concentrations, water isotopes and sulfate isotope measurements in groundwater and surface water, using reciprocal [$SO_4$] vs. $\delta^{34}S$ diagrams to distinguish agricultural and geological sulfate end-members and indicate their mixing trends, and a $\delta^2H$ vs. $\delta^{18}O$ plot to identify groundwater recharged since dam construction upstream of the study area. Novel aspects of the study include the use of the water isotopes to distinguish groundwater samples likely to have been affected by intensive agriculture, which developed following dam construction, and the application of a sample set to a complex study area. A full understanding of salinity sources in surface water and groundwater in the study area is important for future decisions on environmental management in that area and beyond; incorrect or inadequate conclusions drawn from the scientific literature could lead to waste of effort and resources in attempts to mitigate alleged agricultural salinity sources that are in fact geological.

### 1.1. Study Area

The Hueco Bolsón and its northern extension into Tularosa Valley, along with Mesilla Valley-Jornada del Muerto basin (Figure 1a), are large elements of the southern part of the Neogene Rio Grande Rift. The Rift in this area comprises a set of deep alluvial basins separated by hard-rock mountain ranges and is an extensional tectonic feature resembling the Basin and Range Province [20] but may be a separate entity [21]. The Rio Grande leaves Mesilla Valley through a hard-rock barrier between the Franklin Mts. and the Sierra de Juárez, entering the Hueco Bolsón at El Paso and Ciudad Juárez. The river flows along the southern end of the Hueco Bolsón to the basin terminus between Fort Quitman and Indian Hot Springs, marking the border between the USA and Mexico in that area (Figure 1b).

The climate is temperate and arid; at El Paso during the period 1942–2016, average maximum and minimum temperatures were 25 °C and 10 °C, respectively, and average annual rainfall was 217 mm [22]. Precipitation may occur throughout the year; 62% falls between June and September when the North American Monsoon is active. Tropical depressions provide heavy rain in September and October in some years.

The evolution of the Neogene basin fill in the Hueco Bolsón reflects the relationship between the basin and the Rio Grande. In the Miocene and Pliocene, the river fed a terminal lake in which clay and evaporite accumulated. During the Pleistocene, internal drainage was replaced by the present configuration in which the upper and lower sections of the Rio Grande became integrated. Initially, the river entered the Hueco Bolsón north of the Franklin Mts. And flowed along the eastern flank of the Franklin Mts. and the Sierra de Juárez [23,24]. Since ca. 0.67 Ma, the river has entered the Hueco Bolsón south of the Franklin Mts. Basin fill locally exceeds a thickness of 2700 m [25], and is regionally termed the Santa Fe Group (SFG) with upper, middle and lower divisions [26]. The upper, fluvial or alluvial unit of the SFG is locally termed the Camp Rice Formation, and the middle and lower, mainly lacustrine units are termed the Fort Hancock Formation [27]. The Rio Grande Alluvium, consisting of poorly

consolidated sand and gravel of Holocene or late Pleistocene age, underlies the flood plain of the river to a maximum depth of 60 m. Figure 2 is a cross-section of the basin fill in part of the study area.

In 2020, the floodplain and its surrounding area were home to a population, mainly urban, including 865,657 in El Paso County, 1,512,450 in Ciudad Juárez and 20,000 in rural parts of the Valle de Juárez [28,29]. Municipal water supply derives mostly from groundwater. Agriculture is almost entirely limited to the Rio Grande floodplain. Principal activities include raising of cattle and crops including forage, cotton, pecan nuts and vegetables. A wide variety of soil amendments is used on irrigated land [10] for fertilization, sources of Fe, and adjustment of soil pH and wetting properties. Many amendments contain high sulfate concentrations, e.g., ammonium sulfate, sulfuric acid, Ca-Fe sulfate mixtures and gypsum.

### 1.2. Basin Hydrology

The Rio Grande Aquifer is a shallow, brackish aquifer within the Rio Grande Alluvium, extending beneath the floodplain to a maximum depth of 30 m [30]. The floodplain is supplied with river water from reservoirs in New Mexico via a system of canals, and a secondary system of canals ("drains") removes shallow groundwater from cultivated areas. The deeper, regional Hueco Bolsón Aquifer occurs in the SFG throughout the basin; the potentiometric surface indicates inflow from the Tularosa Valley to the north under pre-development and subsequent conditions [31,32]. The Hueco Bolsón Aquifer discharged vertically into the Rio Grande Aquifer under pre-development conditions, a pattern that has reversed as a result of pumping [30,33]. The Pleistocene Rio Grande paleochannel strongly influences groundwater flow. East of the Franklin Mts. it conveys fresh water that mainly originated from the Organ Mts. [32]. Beneath Ciudad Juarez, the paleochannel is recharged from the present course of the river [34]. Groundwater in the central basin, north of the present course of the river, has high salinity from contact with salty lacustrine sediments [11].

### 1.3. Previous Isotope Studies, Hueco Bolsón

Studies of stable O and H isotopes showed that the Rio Grande surface water follows a single, clearly-defined evaporation trend from the river source in Colorado to the Hueco Bolsón [14]. Groundwater recharged by the river is therefore readily distinguished from native Hueco Bolsón recharge [19,34–36]. Native groundwater plots along the global meteoric water line, GMWL [37] or along evaporation trends different from those of the river. River-derived groundwater can be further distinguished according to whether recharge occurred before (less evaporation) or since (more evaporation) the construction of Elephant Butte Dam in New Mexico in 1916. This distinction is expressed in isotope layering of groundwater beneath the river in El Paso and provides a local means for constraining groundwater residence time [36]. The distinction is useful for the present study because agricultural use of Rio Grande water in the Hueco Bolsón has increased greatly since water from the Elephant Butte reservoir became available.

Sulfate in central basin (CB, see Figure 1) and mountain front (MF) groundwater from the Hueco Bolsón north of the Rio Grande consists of mixtures of three probable end members: Permian marine gypsum supplied from Tularosa Valley ($\delta^{34}$S +10 to +12‰), igneous sulfide from the Organ Mountains (−2‰) and atmospheric dust (+7 to +8‰) [11]. A few shallow CB groundwater samples showed evidence of bacterial sulfate reduction; these samples are also richer in Br than most deeper CB groundwater. Pre-dam groundwater beneath Ciudad Juárez contains mixtures of river-derived sulfate like that at present observed in the Rio Grande between Albuquerque, New Mexico and Elephant Butte reservoir with sulfate from native Hueco Bolsón groundwater. Conversely, sulfate in post-dam groundwater resembles that in present-day Rio Grande surface water reaching the Hueco Bolsón [19].

## 2. Materials and Methods

Groundwater (48 samples), surface water (52), mineral samples (5) and soil amendments (2) were collected in or adjacent to the Rio Grande floodplain between Sunland Park, New Mexico, and Indian Hot Springs, Texas. Groundwater samples were collected from wells in continual use, without further purging. Detailed locations are given in Supplementary Material: Table S1 and Figures S1–S3. Sulfate isotope and anion data for these samples are previously unpublished; values of $\delta^2H$ and $\delta^{18}O$ were measured where necessary to supplement previously published data. We also consider numerous data from the literature as shown in Table S1. Water samples were stored in sturdy plastic bottles. No preservative was added; nitrate as a suppressant of bacterial sulfate reduction is undesirable because it can lead to erroneous measurements of $\delta^{18}O_{SO4}$. No samples were subsequently found to have developed $H_2S$ smell. Sulfate was extracted from filtered solutions as $BaSO_4$ by addition of excess $BaCl_2$ solution at pH $\leq$ 2. Precipitate was collected by filtration, washed and dried. Stable S isotopes were measured on 1 mg aliquots of $BaSO_4$. Sulfur was converted to $SO_2$ in a Costech® elemental analyzer and measured using a Thermo Electron Delta Plus XL® (Bremen, Germany) continuous flow mass spectrometer. For O isotopes, 1 mg $BaSO_4$ was analyzed using a continuous-flow isotope ration mass spectrometer (Finnigan Delta X Plus, Bremen, Germany) coupled with a thermal combustion elemental analyzer. Standardization is based on international standards OGS-1 and NBS123, and several other sulfide and sulfate working standards that have been compared between laboratories. Values of $\delta^{34}S$ and $\delta^{18}O_{SO4}$ are reported relative to Vienna Canyon Diablo Troilite (VCDT) and Vienna Standard Mean Ocean Water (VSMOW), with analytical precisions of 0.13‰ (Iσ) and 0.9‰ (Iσ), respectively, according to repeated analysis of laboratory standards. One standard was run with every ten unknowns.

Stable O and H isotopes were measured using a Finnigan Delta S dual-inlet gas-source isotope ratio mass spectrometer relative to international reference materials VSMOW and SLAP. For hydrogen, water was reacted at 750 °C with Cr metal using a Finnigan H/Device. For oxygen, water was equilibrated with $CO_2$ gas at approximately 15 °C in an automated equilibration device. Analytical precision (1σ) is 0.9‰ or better for $\delta^2H$ and 0.08‰ or better for $\delta^{18}O$ on the basis of repeated internal standards. One standard was run with every ten unknowns.

Anion analyses on samples collected by the authors were performed by ion chromatography in the University of Arizona SAHRA Hydrochemical Laboratory and later in the Hydrogeology Laboratory at California State University-Los Angeles following guidelines in EPA Method 300.0 [38]. Analytical precision (1σ) was 5% of the analysis, or better. Where anion data from other sources are used, the methods were reviewed to try to determine if standard methods and procedures were used.

Relationships between $\delta^{34}S$ and $[SO_4]$ are presented graphically on plots of reciprocal concentration (in this instance $100/[SO_4]$) vs. $\delta^{34}S$, in which mixing trends appear as straight lines.

## 3. Results

Surface water and groundwater data from the 2001–2005 sampling campaigns are listed in Supplementary Material, Table S1. Anion concentrations ($[SO_4]$ and chloride $[Cl]$) along with $[SO_4]/[Cl]$ of all Rio Grande surface water samples used in this study are shown as a function of distance from the river source in Figure 3. Because samples were taken under a variety of flow conditions, anion concentrations show considerable scatter. Maximum concentrations and $[SO_4]/[Cl]$ increase downstream as far as km 1150.

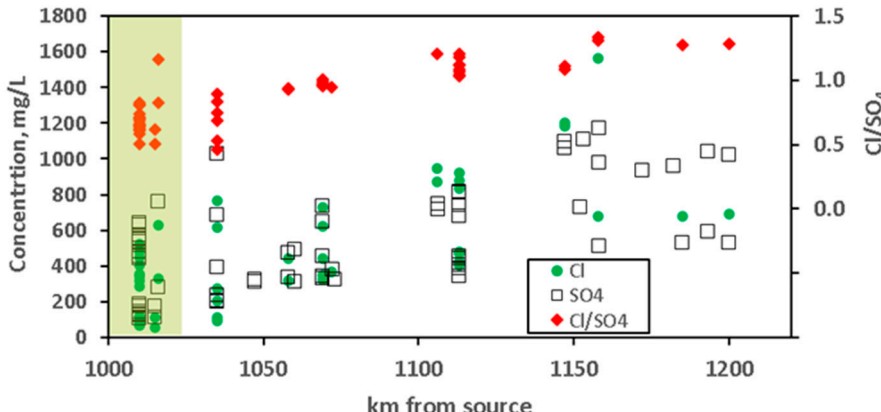

**Figure 3.** Plots of [SO$_4$], [Cl] and [SO$_4$]/[Cl] for all Rio Grande surface water samples used in this study, as a function of location indicated as km from the river source. Samples are from the terminus of Mesilla Valley (green shading) and the Hueco Bolsón (white area).

*3.1. Groundwater*

For the purposes of this study, we classify groundwater samples as follows, on the basis of hydrogeology of the basin with insights from $\delta^{18}$O and $\delta^2$H data. Groundwater originating as recharge from the Rio Grande occurs in the Rio Grande Aquifer and in the Camp Rice Formation beneath Ciudad Juárez. These samples plot on the Rio Grande evaporation line (RGEL) [14] and fall into two distinct groups (Figure 4): pre-dam water ($\delta^{18}$O < −10‰) and mixtures containing post-dam water ($\delta^{18}$O > −10‰), termed post-dam below. The isotope distinction results from intense evaporation in large reservoirs upstream of the study area, mainly at Elephant Butte [14] where a dam completed in 1916 impounds several years' flow of surface water. The distinction is consistent with $^{14}$C and $^3$H data in groundwater beneath Ciudad Juárez [19,34] and cannot be explained by alternative hypotheses [34]. Post-dam groundwater overlies pre-dam groundwater beneath the Rio Grande floodplain in El Paso [36]. The isotope distinction extends to the floodplain throughout the study area. Pre-dam and post-dam groundwater in the floodplain occupy different fields in Figure 5; post-dam water generally has higher [SO$_4$] than pre-dam water, and ranges of $\delta^{34}$S overlap. Two outliers of pre-dam water with negative values of $\delta^{34}$S are from Ciudad Juárez and may represent oxidation of sedimentary sulfide [19].

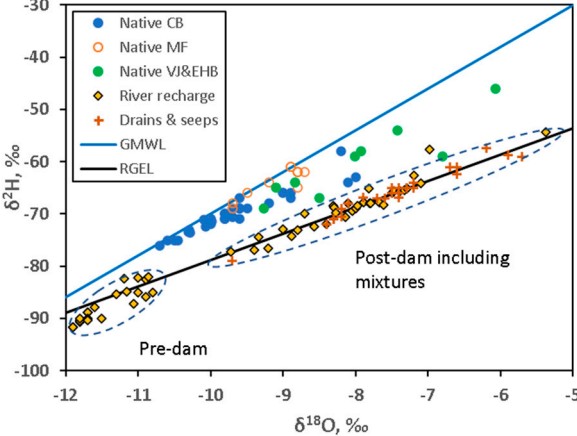

**Figure 4.** $\delta^2$H vs. $\delta^{18}$O for all groundwater and surface water samples from the Hueco Bolsón used in this study. CB = central basin; MF = mountain front; VJ = Valle de Juárez; EHB = east Hueco Bolsón; GMWL = global meteoric water line [37]; RGEL = Rio Grande evaporation line [14].

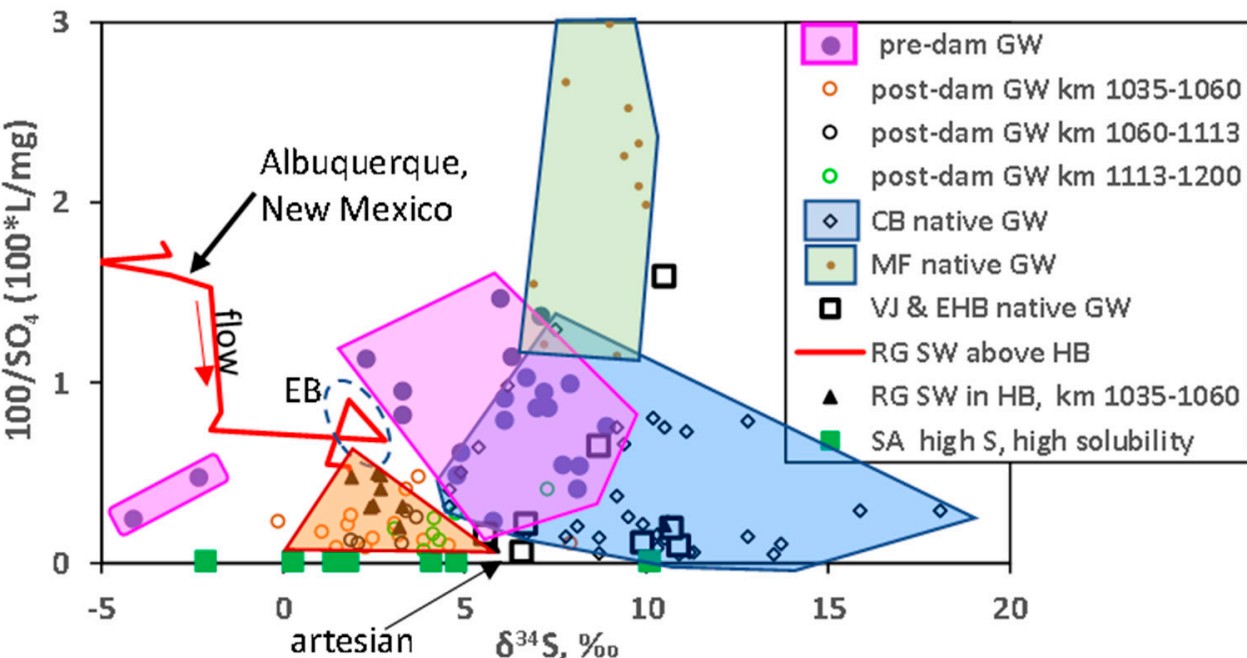

**Figure 5.** Reciprocal sulfate concentration vs. $\delta^{34}S$ for Hueco Bolsón (HB) groundwater (GW), in relation to Rio Grande (RG) surface water (SW) upstream of the Hueco Bolsón. Groundwater from the Rio Grande floodplain, including Ciudad Juárez, is distinguished as "pre-dam" and "post-dam" according to distinctions in $\delta^{18}O$ and $\delta^{2}H$ arising from the impoundment of river water by Elephant Butte (EB) Dam, New Mexico, completed in 1916 (see Figure 4 and [34]). The brown triangle includes most post-dam samples. CB = central basin; MF = mountain front; VJ = Valle de Juárez; EHB = east Hueco Bolsón, SA = soil amendments, plotted for an assumed sulfate concentration of 0.1 m. Data sources: CB and MF: [11]; RG flood plain (pre-dam and post-dam): [18,19], this study; RG: [14], this study; SA: [7], this study.

Groundwater originating as recharge of native Hueco Bolsón surface water plots on the GMWL at $-11 < \delta^{18}O < -9$‰, or on evaporation trends originating in that interval and distinct from the RGEL (Figure 4). Mountain front (MF) groundwater from the Rio Grande paleochannel on the east flank of the Franklin Mts. is largely distinct from adjacent central basin (CB) groundwater in [SO$_4$]. Figure 5 shows relationships between $\delta^{34}S$ and reciprocal [SO$_4$] for groundwater. CB groundwater has a broad range of $\delta^{34}S$ (+5 to +18‰) compared with MF groundwater (+7 to +10‰). Groundwater to the southeast (EHB and VJ areas, Figure 1b) ranges widely in [SO$_4$] and includes sulfate-rich water with a $\delta^{34}S$ range of +5 to +11‰. Two samples from artesian wells near El Millón gave $\delta^{34}S$ values of +5.6 and +6.7‰.

Values of $\delta^{18}O_{SO4}$ are available for CB groundwater (+6 to +15‰, [11]) and a small set of pre-dam groundwater samples (+6 to +15‰, this study).

### 3.2. Irrigation Drains and Seeps

The samples, which represent shallow groundwater discharging from irrigated fields, were taken from drains serving the area between Fabens and Fort Hancock (corresponding approximately to 1069–1113 km along the river). All have $\delta^{18}O$ and $\delta^{2}H$ values corresponding to post-dam groundwater (Figure 4). In most cases, $\delta^{34}S$ values are between 0 and +6‰, overlapping the ranges for post-dam groundwater and most soil amendments (Figure 6). Several samples have higher sulfate than post-dam groundwater. Two outlier samples with $\delta^{34}S$ between +9 and +10‰ were collected at Caseta (km 1113) and near Tornillo (km 1080).

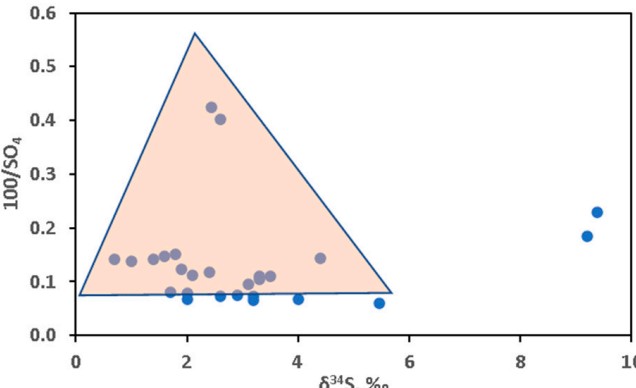

**Figure 6.** Reciprocal sulfate concentration vs. $\delta^{34}S$ for agricultural drains and seeps between Fabens and Fort Hancock. The brown triangle corresponds to the field of post-dam groundwater in Figure 5.

### 3.3. Rio Grande Surface Water

Samples from the Rio Grande in the Hueco Bolsón were taken at high and low flow conditions in order to represent a wide range of [SO$_4$]. Where multiple measurements are available for a single site, concentrations of sulfate and chloride show wide ranges (Figure 3), reflecting a variety of flow conditions in the river. In general, maxima and minima for both anions increase downstream as far as km 1158. Ratios of [Cl]/[SO$_4$] increase downstream to km 1058.

Three linear mixing trends are apparent in a plot of reciprocal sulfate concentration vs. $\delta^{34}S$ (Figure 7). Trend a is defined by samples between 1037 and 1060 km, trend b for 1068 to 1113 km, and trend c for 1147 to 1200 km. Note that the sense of each trend indicates increasing [SO$_4$], not distance along each interval of the river. For each trend, a high-sulfate end-member (A, B, C, respectively) indicates a likely composition of saline water supplied to the riverbed.

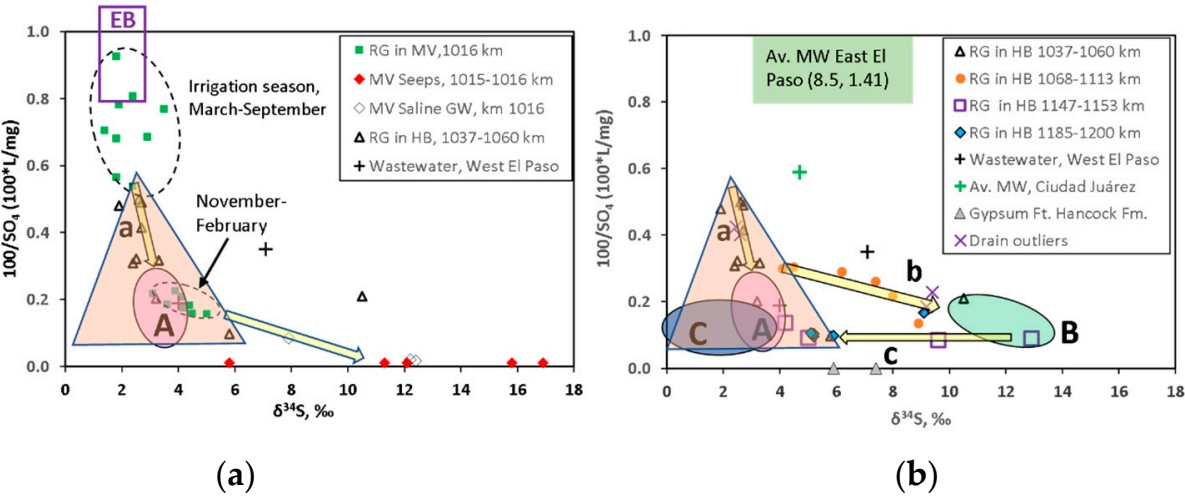

(**a**)  (**b**)

**Figure 7.** Reciprocal sulfate concentration vs. $\delta^{34}S$ for Rio Grande (RG) surface water: (**a**) for surface water entering the Hueco Bolsón (HB) from Mesilla Valley (MV). (**b**) for the river in the Hueco Bolsón. HB, MV = Rio Grande water from the Hueco Bolsón, Mesilla Valley respectively. River locations shown as km from the source. Block arrows indicate mixing trends a, b and c (see text) and sulfate mixing in non-irrigation season surface water at km 1016 near the terminus of Mesilla Valley; A, B and C are proposed end member compositions for mixing trends a, b and c, respectively. Av. MW = arithmetic average of data for municipal supply wells sampled in this study. Data for Mesilla Valley (Rio Grande surface water, salt seeps and wastewater in West El Paso) are from [7,39]. Other data are from this study and [11,18].

Surface water data for site 1016 km near the terminus of Mesilla Valley [10] are shown for comparison (Figure 7a). These data fall into two groups, April-October (irrigation season, with high flow, lower [SO$_4$] and lower $\delta^{34}$S, and November-March (non-irrigation season, low flow, higher [SO$_4$] and higher $\delta^{34}$S). Data for Rio Grande surface water between northern New Mexico and Mesilla Valley were measured during this study on samples taken for [14]. Treated wastewater is discharged at times into the riverbed. Data for West El Paso (upstream of the El Paso Narrows) are from [10]. For East El Paso and Ciudad Juárez, no direct measurements on wastewater are available; the points labeled municipal wells (MW) in Figure 7b are averages of all supply wells sampled [19,34]. All three points are distinct from the river dataset.

Values of $\delta^{18}$O$_{SO4}$ in Rio Grande surface water fall within a narrow range, +7.9 to +10.1‰ (Figure 8), compared with +5.1 to +7.9‰ for river water leaving Mesilla Valley [10].

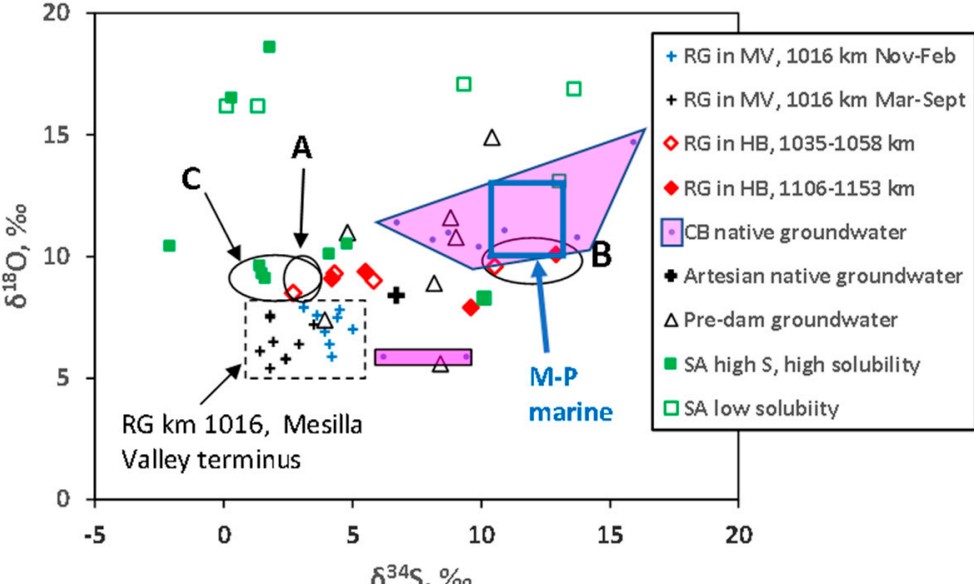

**Figure 8.** Plot of $\delta^{18}$O$_{SO4}$ vs. $\delta^{34}$S, showing available data for Rio Grande, RG, surface water and groundwater in the Hueco Bolsón (HB) from this study and [11,18] in relation to data at km 1016 near the terminus of the Mesilla Valley (MV) from [10] (site 20) and high-sulfur soil amendments (SA) from this study and [10]. RG = Rio Grande (locations shown as km from the source); CB = Central Basin; M-P marine = Middle Permian marine sulfate [40]. A, B and C correspond to end-members shown in Figure 7.

*3.4. Soil Amendments*

The data plotted in Figures 5, 7 and 8 are mainly for high-sulfate amendments from [10]. To these are added a single measurement each on sulfuric acid, with ($\delta^{34}$S, $\delta^{18}$O$_{SO4}$) = (+1.8, +18.2‰) and gypsum (+13.0, +13.1‰). The entire dataset has ranges of +2 to +17‰ for $\delta^{34}$S and +9 to +18‰ for $\delta^{18}$O$_{SO4}$.

**4. Discussion**

*4.1. Pre-Dam Versus Post-Dam Groundwater*

In Figure 5, the field of pre-dam groundwater largely falls between present-day Rio Grande surface water upstream of Elephant Butte Dam and CB native groundwater. Sulfate in such samples therefore represents mixtures of sulfate from pre-dam river water and CB water [19]. Post-dam groundwater contains higher [SO$_4$] than pre-dam groundwater. Most post-dam groundwater sulfate consists of mixtures (brown triangle in Figure 5) between sulfate from surface water entering the basin and sulfate with $0 < \delta^{34}$S $< +6$‰ that is consistent with the $\delta^{34}$S range of most high-sulfur soil amendments in use in the basin (Figure 8). High [SO$_4$] in post-dam groundwater, relative to pre-dam groundwater,

therefore results from use of soil amendments. This conclusion is supported by the similar range of $\delta^{34}S$ in drain water (Figure 6), which represent groundwater most likely to be affected by soil amendments. Drain water has $[SO_4]$ at the high end of the range found in post-dam water sampled from wells, because production from supply and irrigation wells is biased towards better-quality, low-salinity water.

### 4.2. Rio Grande Surface Water Entering the Hueco Bolsón

Multiple river water samples from [10] at km 1016 near the terminus of Mesilla Valley (Figure 9) fall into two groups (Figure 7). Between March and September, which is irrigation (high-flow) season, river water is dominated by low-$[SO_4]$ releases from Elephant Butte reservoir. From November to February (low-flow season), river water is high in $[SO_4]$ and influenced by seepage of deep-basin, saline groundwater into the riverbed at the terminus of Mesilla Valley. These samples form a linear trend indicating mixing with high-$\delta^{34}S$ sulfate like that in salty seeps and salt crusts at the basin terminus.

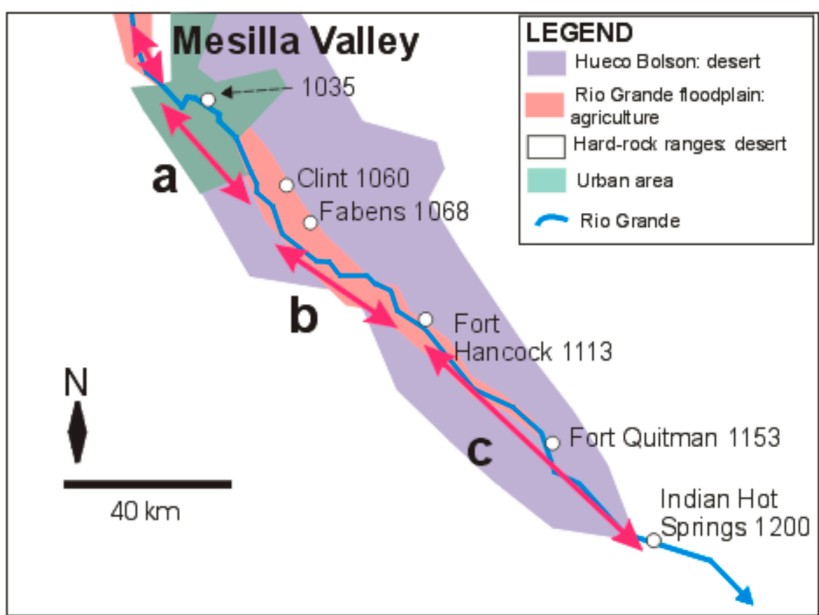

**Figure 9.** Map of the study area showing the river reaches in which of mixing trends a, b and c (Figure 7) occur, relative to land use zones. Numbers such as 1035 indicate km from the river source.

### 4.3. Rio Grande Surface Water, 1035–1060 km

River water samples in the Hueco Bolsón from El Paso to Clint form a mixing trend, a, between high-flow and low-flow river water leaving Mesilla Valley, the low-flow end member A corresponding to the lower-$\delta^{34}S$ end of the range for river water in non-irrigation season (Figure 7a). Additions of sulfate from soil amendments are possible near km 1060, but unlikely at km 1035 (El Paso) where most of the samples were collected, and where the floodplain has been occupied by urban development for many decades (Figure 9).

### 4.4. Rio Grande Surface Water Downstream of Fabens

The section of the river between Fabens and Fort Hancock (1068–1113 km), shows a different mixing trend, b, requiring a high-$[SO_4]$ end member B (Figure 7) with $\delta^{34}S$ values exceeding +9‰. The highest-$\delta^{34}S$ sample, +13‰, in the section between Fort Hancock and Fort Quitman (sampled at 1147–1153 km) probably represents this end member, which may contain reworked Permian marine sulfate. Such sulfate is plentiful among native basin groundwater samples from north and south of the river (compare Figure 5). The two outlier samples from drains fall on trend b. The third mixing trend, c, between Fort Hancock and Indian Hot Springs (1147–1200 km), indicates an end member C with $\delta^{34}S \leq +4‰$,

plotting within the field of post-dam groundwater. In this case, mixing may be occurring at constant or decreasing [$SO_4$] (compare Figure 3). Additions of sulfate from gypsum in the Fort Hancock Formation near Fort Quitman are possible, but not sufficiently low in $\delta^{34}S$ (according to the two measurements available) to account for the lower $\delta^{34}S$ values of trend c. End member C is detected towards the basin terminus, where discharge from the Rio Grande Aquifer (i.e., mainly post-dam groundwater in which sulfate derives partly from soil amendments) into the riverbed is likely.

### 4.5. Spatial Distribution of $\delta^{34}S$ in Surface Water

End member B is detected in the riverbed at the broad southern terminus of the Hueco Bolsón-Tularosa Basin, where discharge from the regional aquifer system is likely, and where artesian conditions exist locally. End member C is detected towards the terminus of the floodplain against horsts of Cretaceous sedimentary rock between Fort Quitman and Indian Hot Springs, where the Rio Grande Aquifer (i.e., mainly post-dam groundwater in which sulfate derives partly from soil amendments) is likely to discharge into the riverbed. A variety of mixtures of B and C occur in this area, indicating that one or the other source of sulfate dominates discharge locally. At Indian Hot Springs, km 1200, end member C is the dominant sulfate source. A single sample from 1037–1060 km that plots with trend b, and the discharge of both of B and C downstream of Fort Quitman indicates local complexity in the spatial pattern of supply of sulfate-rich water to the riverbed.

### 4.6. Sulfate-$\delta^{18}O$ as a Constraint

Rio Grande river water in the Hueco Bolsón has a narrow range of $\delta^{18}O_{SO4}$, about +8 to +10‰, which is adopted as the range for end members A, B and C plotted in Figure 8. The 1σ analytical precision of these measurements is 0.9‰; therefore, A overlaps with the field of low-flow season river water from Mesilla Valley, and B with native CB groundwater and with regional Middle Permian marine gypsum. C overlaps with several points for soil amendments. The constraints from the $\delta^{18}O_{SO4}$ data are therefore consistent with the origins of A, B and C determined from the $\delta^{34}S$ and [$SO_4$] data.

### 4.7. Agricultural or Geological Sulfate Sources?

At issue are the sources of sulfate to Rio Grande surface water and to shallow, post-dam groundwater beneath the river in the Rio Grande alluvial aquifer. Sulfate in post-dam groundwater is a mixture of sulfate supplied from Mesilla Valley (itself a combination of agricultural [10] and geological (Figures 5 and 7a) sources) with sulfate added in the Hueco Bolsón from soil amendments. Sulfate in river water between El Paso and Clint, km 1060, is also supplied from Mesilla Valley. This section of the river is a losing reach because of intensive pumping from the subjacent aquifer.

Between Fabens and Fort Hancock, sulfate from native Hueco Bolsón groundwater is discharged into the riverbed and is most easily observed at times of low flow. Such sulfate appears to originate as Permian marine sulfate, which is abundant in the ranges surrounding Tularosa Valley, whence groundwater flows south into the Hueco Bolsón. Permian marine strata are also prominent in mountain ranges and at depth in the southeastern part of the study area [41]. A possible objection to this interpretation might be that Permian gypsum is used as a soil amendment in the Rio Grande floodplain; one such soil-amendment gypsum, with $\delta^{34}S$ = +13.0‰ and $\delta^{18}O_{SO4}$ = +13.1‰ (Figure 8), was sampled for this study. This is an unlikely explanation of end member B (Figure 7b) because it would require that soil amendment sulfate entering the river originate only as this kind of gypsum, where soil amendments of lower $\delta^{34}S$ value are also used, and clearly dominate post-dam groundwater throughout the floodplain.

*4.8. Limitations of Study*

This investigation of sulfate geochemistry constrains sulfate contributions to salinity of surface water and groundwater in the Hueco Bolsón. It does not constrain the contributions of halides. Soil amendments, implicated at least locally as sulfate sources in this study, contain almost no Cl, yet [Cl]/[SO$_4$] increases downstream in river water in the samples used for this study (Figure 3). Separate approaches, like those applied in basins upstream of the study area [14], are required in order to constrain the sources of halides in the Hueco Bolsón.

**5. Conclusions**

Sulfate isotopes, interpreted in conjunction with water isotopes and anion concentrations, are useful for distinguishing agricultural and geological sources of sulfate in the study area. The isotope geochemistry is complex in detail, so that a detailed sample set is required for such an interpretation. Specific conclusions from this study include:

1. Sulfate entering the Hueco Bolsón in Rio Grande surface water is controlled by the interaction of the river with Mesilla Valley agricultural and geologic salinity sources, as described in [14]. Deep-basin groundwater seeps at the terminus of Mesilla Valley contribute measurable sulfate at times of low flow in the river.
2. Overlapping ranges of $\delta^{34}$S are observed in pre-dam (+2 to +9‰, two outliers < 0‰) and post-dam (0 to +7‰) groundwater. Most post-dam groundwater has higher [SO$_4$] than pre-dam groundwater. High [SO$_4$] in post-dam groundwater results from additions of sulfate from common soil amendments.
3. The difference between sulfate in pre-dam and post-dam groundwater (identified by evaporation effects on O and H isotopes) reflects intensification of agriculture in the Hueco Bolsón since the construction of dams and irrigation schemes.
4. Most water from agricultural drains and seeps is post-dam recharge with a $\delta^{34}$S range of 0 to +6‰, corresponding to the range for common soil amendments.
5. In the reach of the river from El Paso to Clint, river processes in Mesilla Valley control sulfate mixing, but soil amendment sulfate may also be present near Clint.
6. In the reach from Fabens to Fort Hancock, discharge of native, saline groundwater with $\delta^{34}$S > +10‰, probably originating as Middle Permian marine sulfate, contributes to high sulfate concentrations in surface water. The mixing trend for such samples includes two drain sample outliers.
7. In the reach from Fort Quitman to Indian Hot Springs, both native saline groundwater and river-derived groundwater discharge to the riverbed. Near Indian Hot Springs, the isotope signature of $\delta^{34}$S is dominantly like that of soil amendments.
8. The $\delta^{18}$O$_{SO4}$ dataset confirms sulfate origins indicated by the larger $\delta^{34}$S dataset.

**Supplementary Materials:** The following supporting information can be downloaded at: https://www.mdpi.com/article/10.3390/hydrology9060095/s1, Table S1: Anion concentration and isotope data for samples used in this study. Figures S1–S3: Detailed sample location maps.

**Author Contributions:** Conceptualization, C.E. and B.H.; methodology, C.E. and B.H.; validation, C.E. and B.H.; formal analysis, C.E. and B.H.; investigation, C.E., B.H., M.M. and J.D.; resources, C.E. and B.H.; data curation, C.E. and B.H.; writing—original draft preparation, C.E.; writing—review and editing, B.H., M.M. and J.D.; visualization, C.E.; supervision, C.E. and B.H.; project administration, C.E. and B.H.; funding acquisition, B.H. All authors have read and agreed to the published version of the manuscript.

**Funding:** This research was funded by SAHRA (Sustainability of semi-Arid Hydrology and Riparian Areas) under the STC Program of the National Science Foundation, Agreement No. EAR-9876800, and by CEA-CREST (Center for Environmental Analysis, Centers for Research Excellence in Science and Technology) under National Science Foundation Cooperative Agreement No. HRD-0317772.

**Institutional Review Board Statement:** Not applicable.

**Informed Consent Statement:** Not applicable.

**Data Availability Statement:** All data used in this article are available in Table S1: Location, anion concentration and isotope data for samples used in this study.

**Acknowledgments:** The authors thank Alfredo Granados Olivas of the Universidad Autónoma de Ciudad Juárez, who organized access to wells in Mexico. They acknowledge well owners who kindly provided access and samples, including El Paso Water Utilities, the United States Geological Survey, Junta Municipal de Agua y Sanamiento de Ciudad Juárez, and numerous private owners. Two anonymous reviewers are thanked for suggesting improvements to the manuscript.

**Conflicts of Interest:** The authors declare no conflict of interest.

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
