# Peer review of "Origins of Sulfate in Groundwater and Surface Water of the Rio Grande Floodplain, Texas, USA and Chihuahua, Mexico"

_hydrology, doi:10.3390/hydrology9060095_

Round 1

Reviewer 1 Report

Dear authors, the paper entitled "Origins of salinity in groundw ater and surface water of the Rio  Grande floodplain, Texas, USA and Chihuahua, Mexico: the  case of sulfate" concerns an interesting issue for Hydrology readers.

I suggest some moderate revisions to improve the quality of your paper. I attach a file with some specific flaws to be addressed, and here in the following you can find some major notes and suggestions.

In the abstract the claim that water stable isotopes can indicate a groundwater age is a bit exaggerate.

In the introduction the authors started to describe the study area (as a general outline) from the beginning of the text, but no maps are linked to this description, and can be particularly difficult to follow the authors in this part of the text; in addition, the map reported in figure 1 is too general.

I suggest to add a new figure 1, to be cited immediately at the beginning of the introduction, as a general outline of the area (for example, the part a) of the current fig.1, a little bit wider), and to add a new figure 2 (to be cited in the next paragraph) with a more detailed description of the studied area (for example fig. 1b with some indication on topography).

In addition, geological map is lacking, and it is not so simple to read the only geological cross section, legend is very synthetic, the meaning of acronyms in the current figure 2 are not immediately readable); topographic surface is flat? which are USF, MSF, MLSF...?

Geological history description, instead, is very clear.

Moreover, in the introduction you stated that you use extensive dataset of isotopes for soil amendments and surface water. In groundwater, where the samples are from? A close area? or further? And you said that the obtained data are referred to both the agricultural impact and geological effect; it is not clear for me in which way you consider the geological effect starting from the isotopic data; you should clarify this aspect here in the introduction, where you present the goal of this work.

Please better specify the role of dam in aquifers recharge/discharge.

At the beginning of materials and methods section, an introduction reporting where samples have been collected is preferable (number of samples, type of samples and so on); maybe a table can help.

Before the results chapter (in which results are presented), it is not so clear, for me, if stable isotopes analyses of water have been made or only analyses of sulphate. Please be more specific in the text.

Figure 4 and figure 5 (and their description in the text): please clarify on which basis you classify groundwater as pre-dam and post-dam. Is it only a temporal classification? If so, why values have been so modified? It would be very important to discuss this point and give a clarification.'cause this point is a key point even for the discussion of the sulphate isotopes values, presented further in the manuscript.

In the discussion, the presence of higher sulphate after the dam construction is well explained and presented; in general water and groundwater terms are preferred in the singular notation, not plural.

Referring to the final part of discussion, when you stated that "A possible objection to this interpretation might be that Permian gypsum is used as a soil amendment in the Rio Grande floodplain": have you data on isotopic content of such amendments? have you performed some batch tests or lisciviation to extract water and making isotopic analysis?

The conclusion chapter should be more discursive, or at least you should add a brief introduction before the numbered list.

Reviewer 2 Report

  1. The title should be reconsider. Maybe focus on the sulfate rather than salinity. In addition, “groundwater” should not be written as “groundwa” and “ater”.
  2. The first or two sentences in the abstract are suggested to be used for introducing the meaning or background of the present research.
  3. Ln 16-17: we usually use radio isotopes for aging groundwater rather that stable isotopes.
  4. For the abstract section: authors should reconsider what is the novelty and main findings for the present research. The present manuscript is poor and hard to get the novelty.
  5. There are too many keywords. It is not necessary for 4 keywords of place.
  6. The introduction section should be reorganized. Authors are recommended to introduce the background/meaning of the present research in the first paragraph. Then review the research progress in the field over the world. This could use 1-3 paragraphs. Lastly, introduce briefly the typicality of the study area and what would you do in present research. Avoid only focus the local interests. Remember that this is an international journal.
  7. The hydrogeology and geology maps are missing.
  8. The sampling map is also missing.
  9. LN 152-175: the review should not be putted in the study area section.
  10. What is the QA/QC for sample analysis?
  11. What is the precision of the chemical analysis?
  12. Land use and land cover map would be helpful in the discussion.
  13. Spatial map should be involved in section 4.5.
  14. The conclusion should be reorganized. Please refer to published paper how to organize a conclusion section.

Round 2

Reviewer 1 Report

Thank you for your revision. My intention is to specify some suggestions/requests I had made in the previous review round.

In the abstract, you cannot state this sentence without any explanation: "Water isotopes (δ2H, δ18O) indicate groundwater age relative to dam construction.", because the sentence itself, as it's formulated, cannot be considered as a general rule. My suggestion is to change introducing previous studies results and comparison with non-stable isotopes results.

In my opinion, the topography and orography should be indicated in a figure within the manuscript; moving this important information out of the paper certainly not favor the readability of the paper itself.

Author Response

In the abstract, you cannot state this sentence without any explanation: "Water isotopes (δ2H, δ18O) indicate groundwater age relative to dam construction.", because the sentence itself, as it's formulated, cannot be considered as a general rule. My suggestion is to change introducing previous studies results and comparison with non-stable isotopes results.

We have modified the sentence to read:

From previous studies, water isotopes (δ2H, δ18O) in the study area indicate groundwater age relative to dam construction upstream.

In my opinion, the topography and orography should be indicated in a figure within the manuscript; moving this important information out of the paper certainly not favor the readability of the paper itself.

We have added an inset to Fig. 1b, showing topographic contours, and have moved the mountain range names to the inset.

Reviewer 2 Report

The manuscript has been improved and can be considered for accepting.

Author Response

No further changes were requested.